# Endoscopic Vacuum-Assisted Closure (E-VAC) in Septic Shock from Perforated Duodenal Ulcers with Abscess Formations

**DOI:** 10.3390/jcm13020470

**Published:** 2024-01-15

**Authors:** Bogdan Mihnea Ciuntu, Adelina Tanevski, David Ovidiu Buescu, Valerii Lutenco, Raul Mihailov, Madalina Stefana Ciuntu, Mihai Marius Zuzu, Dan Vintila, Mihai Zabara, Ana Trofin, Ramona Cadar, Alexandru Nastase, Corina Lupascu Ursulescu, Cristian Dumitru Lupascu

**Affiliations:** 1Department of General Surgery, Faculty of Medicine, Grigore T. Popa University of Medicine and Pharmacy, 16 Universitatii Street, 700115 Iasi, Romania; buescu_david-ovidiu@d.umfiasi.ro (D.O.B.); zuzumihai@gmail.com (M.M.Z.); danersilia@yahoo.com (D.V.); mihai.zabara@umfiasi.ro (M.Z.); ana-maria.trofin@umfiasi.ro (A.T.); ramona-petronela.cadar@umfiasi.ro (R.C.); nastase_grigorie-alexandru@d.umfiasi.ro (A.N.); cristian.lupascu@umfiasi.ro (C.D.L.); 2General Surgery Clinic, “St. Spiridon” County Emergency Clinical Hospital, 1 Independence Boulevard, 700111 Iasi, Romania; 3Department of General Surgery, Faculty of Medicine, “Dunarea de Jos” University of Medicine and Pharmacy, 800010 Galati, Romania; vl187@student.ugal.ro (V.L.); raul.mihailov@ugal.ro (R.M.); 4General Surgery Clinic, “St. Apostol Andrei” County Emergency Clinical Hospital, Strada Brăilei 177, 800578 Galati, Romania; 5Faculty of Medicine, Grigore T. Popa University of Medicine and Pharmacy, 16 Universitatii Street, 700115 Iasi, Romania; mg-eng-24128@students.umfiasi.ro; 6Department of Radiology, Faculty of Medicine, Grigore T. Popa University of Medicine and Pharmacy, 16 Universitatii Street, 700115 Iasi, Romania; corina.ursulescu@umfiasi.ro

**Keywords:** E-VAC, negative pressure therapy, perforated duodenal ulcer, abscess

## Abstract

This case report underscores the importance of utilizing E-VAC (endoscopic vacuum-assisted closure) in the treatment of a perforated duodenal ulcer complicated by the formation of a subphrenic abscess and septic shock. It showcases how E-VAC can effectively mitigate the risk of further complications, such as leakage, bleeding, or rupture, which are more commonly associated with traditional methods like stents, clips, or sutures. As a result, there is a significant reduction in mortality rates. A perforated duodenal ulcer accompanied by abscess formation represents a critical medical condition that demands prompt surgical intervention. The choice of the method for abscess drainage and perforation closure plays a pivotal role in determining the patient’s chances of survival. Notably, in patients with a high ASA (American Association of Anesthesiologists) score of IV-V, the mortality rate following conventional surgical intervention is considerably elevated. The management of perforated duodenal ulcers has evolved from open abdominal surgical procedures, which were associated with high mortality rates and risk of suture repair leakage, to minimally invasive techniques like laparoscopy and ingestible robots. Previously, complications arising from peptic ulcers, such as perforations, leaks, and fistulas, were primarily addressed through surgical and conservative treatments. However, over the past two decades, the medical community has shifted towards employing endoscopic closure techniques, including stents, clips, and E-VAC. E-VAC, in particular, has shown promising outcomes by promoting rapid and consistent healing. This case report presents the clinical scenario of a patient diagnosed with septic shock due to a perforated duodenal ulcer with abscess formation. Following an exploratory laparotomy that confirmed the presence of a subphrenic abscess, three drainage tubes were utilized to evacuate it. Subsequently, E-VAC therapy was initiated, with the kit being replaced three times during the recovery period. The patient exhibited favorable progress, including weight gain, and was ultimately discharged as fully recovered. In the treatment of patients with duodenal perforated ulcers and associated abscess formation, the successful and comprehensive drainage of the abscess, coupled with the closure of the perforation, emerges as a pivotal factor influencing the patient’s healing process. The positive outcomes observed in these patients underscore the efficacy of employing a negative pressure E-VAC kit, resulting in thorough drainage, rapid patient recovery, and low mortality rates.

## 1. Introduction

The significance of this case report is to illustrate how the use of E-VAC in the treatment of a perforated duodenal ulcer complicated by subphrenic abscess formation outperforms traditional approaches in terms of postoperative complication incidences and mortality rates. Traditional surgical techniques for perforation closure and abscess drainage, such as sutures, stents, and clips, have been documented to carry a risk for postoperative complications (leaks, bleeding, and ruptures) [1]. Due to these complications, secondary and tertiary surgical interventions become necessary, not only adding further physical burden upon the patient but also increasing the risk of infection and even sepsis [2]. As a result, the mortality rate for patients who have undergone these traditional surgical methods remains high, especially when they have poor ASA scores (IV-V) [3]. This case report demonstrates how the use of E-VAC resulted in the constant and relatively rapid recovery of a patient.

The perforation of duodenal ulcers (PDUs) is more common than gastric ulcers, with an incidence higher in men [4]. Transmural perforated ulcerative lesions result in duodenal content oozing into the peritoneal cavity [4,5,6]. This presents an irritating local process contaminated with microbial germs, leading to peritonitis [1,2,5]. Fistulas can form in the subhepatic space, right parietal colic space, pouch of Douglas, and peritoneal cavity. The free air accumulates under the diaphragmatic dome, resulting in a bilateral pneumoperitoneum. The erosive process of PDUs is due to acid hypersecretion, diet, or stress [4,5,6]. More specific factors favoring perforation include stress, physical fatigue, seasonal painful pustules, ulcerogenic medication, abdominal trauma, excessive food intake, surgical instrumental maneuvers, and imaging diagnostics. It must be noted that the epidemiologic nature of a chronic peptic ulcer (CPU) has changed over the years due to easy access to NSAIDs (non-steroidal anti-inflammatory drugs), the overconsumption of alcohol and cigarettes, and the decreased rates of peptic ulcers due to helicobacter pylori infections thanks to the PPI (proton pump inhibitor) medication [4].

The typical presentation of a PDU is that of an acute abdomen, which is a life-threatening condition. The typical clinical picture consists of a sudden onset of intense epigastric pain characterized as stabbing in nature. Additional symptoms include pain exacerbation upon movement, irradiation to the right hypochondrium, sporadic early vomiting of gastric content, and hematemesis. Physical signs are cold sweats, shallow breathing, tachycardia with normal blood pressure, and a transient subsiding of pain, which is worse upon recurrence [1,2,5,6]. Rare cases with more insidious progression have been recorded, where the patient presents to the hospital with symptoms of less severe complications. Therefore, it is paramount that if a patient presents with subclinical features suggestive of a PDU, a computed tomography (CT) scan must be performed to rule out or confirm the presence of a perforation. Additional lab tests can be used to rule out any differential diagnoses [6,7]. 

By the time a clinical examination has been completed, a differential diagnosis based on the history and physical exam should be established. This can be an elaborate list, including but not limited to conditions such as inflammation (acute pancreatitis and cholecystitis), intussusception, volvulus, ischemia, lithiasis, ulcers, hemorrhage, and perforation [1,2,5,6]. At this stage, an upright chest and abdominal X-ray (X-radiation) is a rapid and cost-effective method that can help to narrow down the differential diagnosis but does have a significant false negative rate. Another advantage of obtaining a plain X-ray is that it can help to identify any other comorbidities of the patient [5,6]. The option for imaging with far better accuracy is CT, which can assist in identifying the problem with increased detail [5,6,7]. In the case of an acute abdomen, a surgical emergency, laparoscopy, or exploratory laparotomy is indicated [5,6]. Before these investigations are undertaken, a blood sample (CBC, BMP, LFT, coagulation, amylase, and lipase) should be obtained to evaluate the patient’s general condition and help to identify any parameters that need correcting. Certain results can further aid in confirming or ruling out the diagnosis and, therefore, direct the physician toward the pertinent treatment [2,7].

Currently, several surgical techniques are available, with the choice of minimally invasive approaches increasing due to some distinct advantages [7,8,9,10]. In the case of a small perforation in a stable patient, conservative treatment can be an option [5,6,7,10].

A nasogastric tube is inserted, and the GI tract is allowed to heal. If the patient is infected with helicobacter pylori, triple antibiotic therapy with a PPI should be prescribed [4,7,10]. For larger perforations or situations where the patient’s condition is deteriorating, surgical intervention is the therapy of choice. The surgical approach could either be laparotomy, laparoscopic, or endoscopic [8,9,10]. Laparotomy and laparoscopy incorporate the same direct closure techniques, that of sutures, clips, omental patches, or, recently, the use of hydrogels [8,9,10,11]. 

Endoscopic vacuum-assisted closure (E-VAC) therapy is a new method for repairing the defects of the upper gastrointestinal system with different etiologies. This method has high success rates. E-VAC therapy consists of placing a sponge either within the lumen or within an abscess cavity connected with a nasogastric tube to a negative pressure system, thereby gradually decreasing the cavity size until complete closure [12]. 

The required materials are a double-lumen nasogastric probe, a vacuum S-size kit, surgical instruments for small surgery, and a vacuum pump. Device preparation requires information about the size of the defect to be explored and also provides data for endoscopic guidance in preparing the contact material (foam kit) and sizing the nasogastric tube dimensions to be mounted on the sponge. Obtaining the necessary data for the intervention is achieved by performing a CT scan. The intervention involves a mixed endoscopic-surgeon-anesthetist team, and the therapeutic gesture would take place in a surgery room, with the patient requiring general anesthesia with orotracheal intubation. 

## 2. Results

### Case Report

A 58-year-old male was admitted to general surgery clinic II complaining of superior abdominal pain. Additional symptoms included weight loss, lack of appetite, no bowel movement for 48 h, and a palpable abdominal mass. A physical examination revealed an enlarged abdomen with asymmetry due to a palpable mass of 20 × 20 cm located at the epigastrium and left hypochondrium. Peritoneal irritation signs were negative.

The clinical and paraclinical examinations confirmed a palpable pseudo-tumoral formation at the level of the epigastrium and the left hypochondrium. 

An abdominal-pelvic native CT was performed, which revealed a voluminous collection of mixed hydro-aerial contents and radio-opaque material (an orally administered contrast substance) located in the subphrenic region, extending along the anterior abdominal wall to the supraumbilical region with global dimensions of 98/314/178 mm (AP/T/CC), and was well-defined. A contrasting substance was also present in the GI tract distal to the perforation. Elevation of the right hemi-diaphragm was identified with an overlying lamellar atelectasis. An approximately 19 mm parietal defect was found at the level of the duodenal bulb, with a wide communication along a 100 mm long fistulous tract and a collection adjacent to the gallbladder inferiorly. Peritoneal air bubbles were also present adjacent to the collection. The liver dimensions were increased (right lobe of 183 mm and left lobe of 75 mm anterior-posterior incidence) with a homogenous structure, no dilated extra and intrahepatic biliary ducts, a gallbladder with liquid content with normal wall thickness without radiopaque calculi, a normal permeable portal venous system, and a normal pancreas and suprarenal glands. The spleen dimensions were approximately 132/50/111 mm with a homogenous structure, the kidneys were at the normal positions and had normal dimensions and secretions, and the right superior polar cortical cyst was 8 mm in diameter, with 12 mm lamella of perisplenic fluid an infiltrated aspect of peritoneal fat, a collapsed stomach, and a colon without preparation and without parietal modifications regarding the caliber of a normal size. There was bilateral pleural fluid: 23 mm on the right and 13 mm on the left. There was left basal calcareous pachypleuritis. The urinary bladder was in semirepletion, and the prostate had normal dimensions. The conclusion was a duodenal-covered perforation, subphrenic fistula, and voluminous subphrenic collection predominately full of air and a small quantity of pneumoperitoneum.

Surgical intervention was decided as the therapy. Upon exploratory laparotomy, we revealed the presence of an advanced adhesion process with a mass formation. This mass was found to be compressing the duodenum and pancreas but was not obstructing the GI tract. Once viscerolysis was performed, the presence of a subphrenic abscess located bilaterally containing purulent, aero-digestive content was confirmed. The evacuation of the abscess was performed via the application of three rubber drainage tubes. Because of the important inflammatory status and the high grade of adhesions, the perforation could not be objectified. An endoscopic evaluation was performed, and we found a duodenal defect in segment I, which was communicating with the residual cavity of the abscess (Figure 1).

We decided to use E-VAC. We prepared the required materials as follows: a double-lumen nasogastric tube, an S-sized vacuum kit, small surgery instruments, and a vacuum pump. The sponge was prepared and adjusted to the dimensions of the cavity in question according to the measured dimensions. The nasogastric probe was prepared according to the length of the sponge, with attention placed on the location of the openings through which the secretions would be aspired. The openings were located within the proximal and distal edges of the sponge sleeve. The path of the nasogastric probe was cut in the sponge following an imaginary line through the center along the entire length (Figure 2a,b) (patent number: RO135252). The model of the probe allowed the local instillation of substances such as antibiotics and, at the same time, washing and controlling the tightness of the cavity where it was placed.

The procedure incorporates a multi-departmental team, including an endoscopist, surgeon, and anesthesiologist. The patient requires general anesthesia with orotracheal intubation, and the procedure takes place in an operation room.

Once the preoperative preparation is concluded, an upper digestive endoscopy is performed in order to identify the defect, wash the cavity, and, finally, inspect it. The prepared nasogastric tube is inserted through the nasal orifice and externalized through the oral orifice, at which point the prepared sponge is attached and firmly anchored.

The endoscopist then guides the nasogastric tube with the attached sponge (foam kit) to the level of the defect and places it within the cavity to be drained. The passing of the tube with the sponge into the cavity can be achieved either using via an overtube (a tube with a sponge in front of an endoscope), piggyback (an endoscope following the tube with a sponge), or a flextube (applying the endoscopic forceps a few centimeters from the burette).

The foam kit is visually confirmed to be in place by slightly withdrawing the endoscope, at which time negative pressure aspiration is initiated under visual observation. Once the proper placement is confirmed, the endoscope is fully withdrawn, and the nasogastric tube is firmly fixed to the nasal alar. The negative pressure setting we used was 100 mmHg. Patients with such pathologies who undergo treatment with negative endoluminal pressure are admitted and monitored in the intensive care unit throughout the treatment period.

Bacteriological testing of the peritoneal liquid, including ascites (aerobic culture), revealed candida glabrata and acinetobacter baumannii, and we initiated treatment with imipenem and fluconazole.

The evolution of the patient was positive, and after five days, we changed the vacuum kit under endoscopic review with a decrease in the fistula. The abdominal ultrasound revealed liquid 18 mm to the right of the pleural fine lamellar, and the right diaphragm was mobile without left pleural liquid. There was 58/40 mm of liquid to the right of the subphrenic liquid collection with homogenous content, and the liquid was absent in the Morrison space and under the liver. We performed two more vacuum kit changes at four days, with local and general favorable evolution.

We performed an abdominal-pelvic CT scan, which revealed the closure of the perforation with the absence of extravasation from the gastrointestinal tract of the orally administered contrast. The contrast was present in the jejunum. The right subphrenic collection dimensions were reduced, with mixed air/liquid content at a maximum of 25 mm. At the level of the perforation, anteriorly describes the aspects of a pseudodiverticulum (length 20 mm, permeable lumen 7 mm), which associates with discrete infiltrates of adjacent fat. Bilateral pleural liquid with a thickness of 36 mm on the right and 33 mm on the left was present, with passive atelectasis of the underlying lungs and bilateral basal fibrosis bands [Figure 3].

The E-VAC DI-DII kit was removed on the 21st postoperative day of the second intervention, and the DI defect was found to be closed [Figure 4]. The patient was recovering well with signs of weight gain and, therefore, was determined to be cured and discharged [Figure 5]. 

## 3. Discussion

The strength of this case report lies in that the surgical team involved was experienced with the E-VAC system, having successfully performed similar surgical interventions previously. 

From our data, the endoscopic vacuum system was used for the first time in Romania in the I-II surgery clinic of the “Sf. Spiridon” hospital in Iasi for the treatment of a duodenal fistula that could not be surgically approached in this case. The double-lumen E-VAC system was used for a perforated duodenal ulcer, and the principles and techniques were identical to those of the other perforations within the superior gastrointestinal tract, like esophageal and stomach perforations. A drawback is that there are no recent cases similar in nature that could be used as a control for comparison. 

More case reports and case studies will be required to fully understand the definitive outcomes of the E-VAC system. 

Numerous studies have analyzed the statistically significant differences between laparoscopic and open abdominal surgery. Tulinský L. et al. retrospectively compared the postoperative outcomes of 102 patients requiring intervention for perforated peptic ulcers over six years. According to the obtained data, they observed that the laparoscopic group had a higher incidence of mild complications (27.3% vs. 20.7%), and a higher percentage of uncomplicated postoperative courses (40.9% vs. 24.1%), while the open abdominal group had a higher incidence of patients with severe complications, and the mortality rate was higher (41.4% vs. 13.6%, *p* = 0.001) [13]. The risk factors for predicting postoperative mortality were analyzed by Yalcin M. et al., and they found that an age ≥ 60 years, an ASA score > III, a perforation-surgery interval > 24 h, purulent intraperitoneal contamination, preoperative renal failure, duodenal perforation, and preoperative shock are risk factors affecting mortality [3,14]. Regarding the safety and efficacy of E-VAC systems, Pattynama L. et al. found that E-VAC systems achieve a success rate of 74% (95% CI 57–87%) and an adverse event rate of 5% (95% CI 1–18%) [15]. A meta-analysis conducted by Mandarino FV. et al. compared the results of E-VAC systems versus self-expandable metal stents and found that E-VAC systems have higher success rates (odds ratio [OR] 2.58, 95% CI 1.43–4.66), shorter treatment durations (pooled mean difference −9.18, 95% CI −17.05–−1.32), lower short-term complications (OR 0.35, 95% CI 0.18–0.71), and lower mortality rates (OR 0.47, 95% CI 0.24–0.92) [16].

In this case report, the patient was admitted with a perforated duodenal ulcer and septic shock, providing him with an ASA score of IV. Given these conditions alone, the patient would have had a poor prognosis. Postoperative risk factors for this case were an ASA score > III, duodenal perforation, and preoperative shock. The surgical intervention he received was via an open abdominal approach, which would have increased the mortality rate as well as the risk of severe complications [3,13,14,15,16]. On the other hand, by utilizing the E-VAC system instead of conventional techniques (omental patches and sutures), the outcome and patient recovery were profoundly improved. In patients being constantly monitored in the intensive care unit, with the biological samples being collected daily, we observed an improvement in the inflammatory constants (WBC and PCR) after changing the vacuum kits, these being in close correlation with the cyclicity of the surgical interventions, obtaining the normalization and stabilization of these constants in the last therapy sessions with negative pressure. Studies that referred to the optimal time to keep a sponge at the wound level discussed exchange intervals of up to seven days, while others recommend changing the kit at intervals of three to four days. In our case, we noticed an aggravation of the inflammatory constants starting from the fourth day of keeping the sponge in place, under the conditions in which we used two washing periods within 24 h. Increasing the instillation cycles to more than 1/day led to the delay of the inflammatory response and the possibility of changing the kit even after five days. The patient rapidly recovered without complications and was subsequently discharged.

Statistical studies have demonstrated that different surgical approaches for perforated duodenal ulcers have significant differences regarding recovery, complications, and mortality rates [13,14,15,16]. Upon comparison between conventional approaches (open abdominal) versus minimally invasive (laparoscopic) and between the varying procedures (omental patch, suturing, and E-VAC), our case report demonstrates that the E-VAC system is the option with the best prognosis and lowest mortality rate [17,18,19,20,21,22,23,24,25,26,27]. Without the use of the E-VAC system, this patient with an ASA score of IV did not have a favorable prognosis, having a high risk of postoperative mortality [3,16,17,18,19,20,21,22,23,24,25,26,27,28,29]. Not only was the patient able to recover, but the patient was able to do so without any complications and with rapid recovery. The patient returned to control, which was established 1 month after discharge, with good general condition, digestive tolerance, and physiological intestinal transit. The laboratory analyses carried out during the control do not objectify pathological changes, and the patient continues to be monitored by the gastroenterology service. Therefore, the E-VAC system is a promising new surgical approach for the treatment of perforated duodenal ulcers, providing patients with better prognoses and lower complications and mortality rates [17,18,19,20,21,22,23,24,25,26,27,28,29,30,31,32]. 

Some of the limitations of the method could be summarized as follows: the need for a complex multidisciplinary team consisting of a radiologist, gastroenterologist, surgeon, and intensive care physician who can manage such cases; the time required for the treatment, which often exceeds 40 days; the costs and resources involved in the management of such a case that spends most of its treatment time in the intensive care unit; and the psychological impact on the patient who requires medical care for a long period of time with repeated general anesthesia. Part of the challenges raised with the method are represented by the early recognition of the defect (for the upper digestive tract, this includes esophageal, duodenal perforations, and postinterventional fistulas) in most cases; this occurs after more than 48 h. Other challenges include maintaining the kit in the initial position after extubating the patient (they often become non-compliant with the passage of time); the need for the continuous availability of the team responsible for such interventions in the event of assembly malfunctions during the treatment; mixed teams, reduced in number, willing to approach such a therapy in the management of high digestive perforations.

The application of the treatment for perforations in the upper digestive tract, along with the use of double-lumen probes and the achievement of spectacular results in the healing of patients has led to the proposal of a prototype of triple-lumen probes in clinical practice. This would allow the minimization of the surgical gestures required after the installation of the endoluminal aspiration kit by eliminating feeding jejunostomy; the patient’s nutrition during the treatment would be possible through the lumen specially intended for this procedure, led distally from the lesion. The triple lumen probes proposed by us in the treatment of high digestive perforations bring an advantage in the nutritional management of patients and, at the same time, eliminate the need for feeding jejunostomy.

## 4. Conclusions

Endoscopic vacuum-assisted closure (E-VAC) therapy for the treatment of transmural lesions within the digestive tract is a formidable new option with lower rates of mortality and complications as compared to more traditional surgical techniques. Using previous approaches (like sutures, stents, and clips), the mortality rates of patients with poor ASA scores (IV-V) with complications (such as leakages, fistulas, and abscess formations) are still strikingly high.

The implementation of E-VAC methods has increased in the last two decades with more positive patient outcomes. As demonstrated by our case, our patient was able to fully recover even though he had the usually lethal complications of septic shock and subphrenic abscess formation. Therefore, our conclusion is that E-VAC is a viable mode of treatment for complicated peptic ulcers, with lower mortality rates compared to traditional therapeutic methods.

The promising results obtained in the treatment of upper digestive perforations by our team with the use of double-lumen probes lead us to propose the implementation in the management of such cases of triple-lumen probes that bring benefits by eliminating the need for feeding jejunostomy for patients who require long periods of treatment.

## Figures and Tables

**Figure 1 jcm-13-00470-f001:**
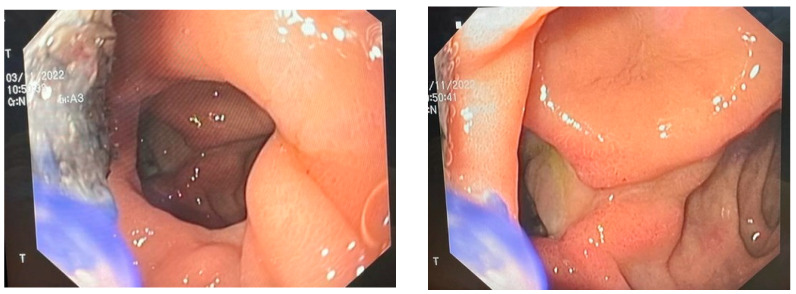
Endoscopic view of the duodenal ulcer.

**Figure 2 jcm-13-00470-f002:**
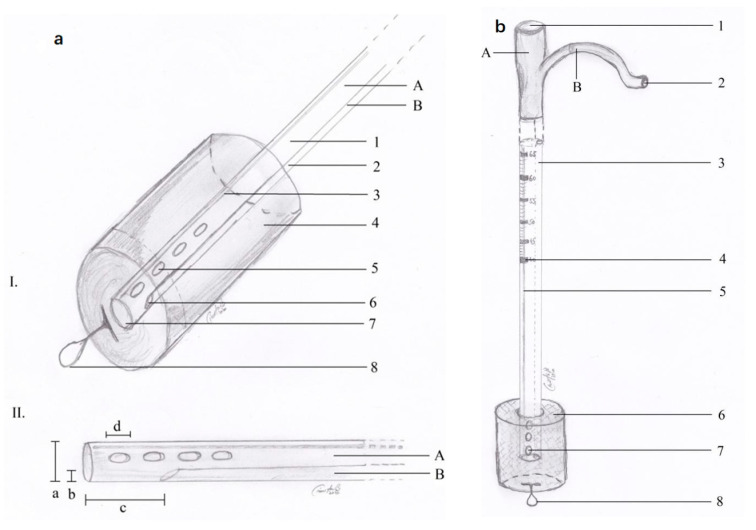
(**a**) E-VAC aspect of the distal end; (**b**) E-VAC device ready for installation.

**Figure 3 jcm-13-00470-f003:**
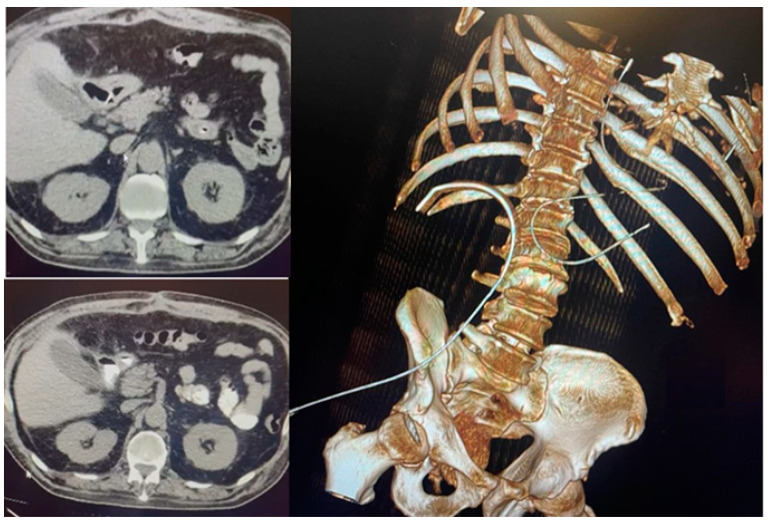
Absence of extravasation from the gastrointestinal tract of the contrast substance.

**Figure 4 jcm-13-00470-f004:**
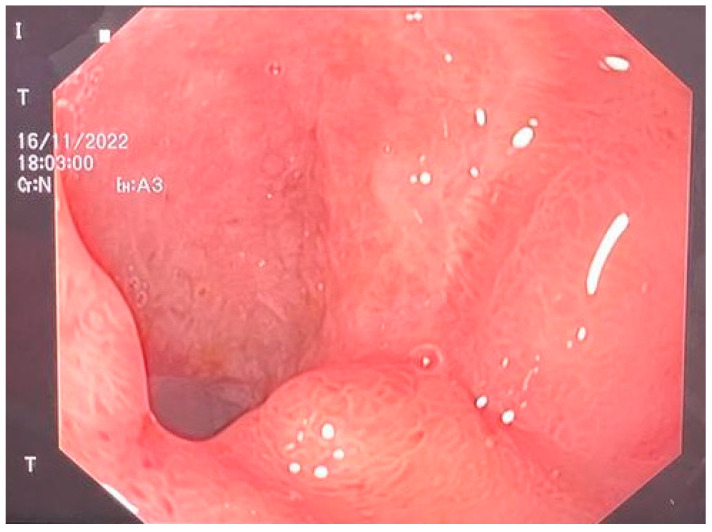
Completely closed duodenum.

**Figure 5 jcm-13-00470-f005:**
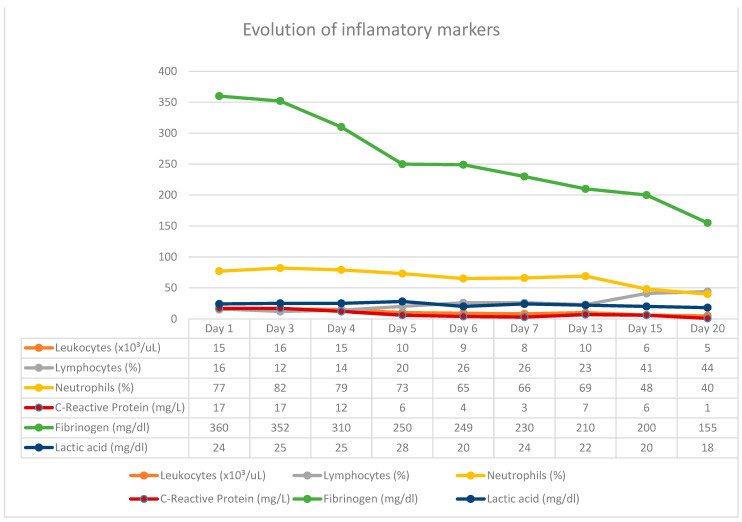
Correlation of inflammatory markers with vacuum kit changes.

## Data Availability

All data generated or analyzed are included in this case report. Further inquiries can be directed to the corresponding author.

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
