# Peer review of "Endoscopic Vacuum-Assisted Closure (E-VAC) in Septic Shock from Perforated Duodenal Ulcers with Abscess Formations"

_jcm, 2024, doi:10.3390/jcm13020470_

Round 1
Reviewer 1 Report
Comments and Suggestions for Authors
- The case report addresses a critical medical condition—perforated duodenal ulcer with complications—which adds value to the medical literature. This report provides a comprehensive overview of the patient's history, symptoms, and clinical presentation, aiding in a thorough understanding of the case.
- Some suggestions
The abstract is extensive and could benefit from concise summarization.
Author Response
we have done the modification that was requested
Reviewer 2 Report
Comments and Suggestions for Authors
The article titled "Endoscopic Vacuum-Assisted Closure (E-vac) in Septic Shock from Perforated Duodenal Ulcer with Abscess Formation" presents a case report highlighting the use of E-vac therapy in treating a perforated duodenal ulcer complicated by a subphrenic abscess and septic shock. The report details the clinical presentation, diagnosis, treatment, and positive outcomes of the patient, emphasising the effectiveness of E-vac in such complex medical conditions. The authors discuss the advantages of E-vac over traditional methods in reducing postoperative complications and mortality rates, particularly in patients with high ASA scores.
Strengths:
- Detailed Case Presentation: The report provides comprehensive details of the patient's condition, treatment process, and recovery, contributing valuable insights into the application of E-vac in complex ulcer cases.
- Clinical Relevance: The discussion of E-vac as an alternative to traditional surgical techniques addresses a critical need in managing complicated ulcer cases, making it highly relevant to current medical practices.
- Supporting Data: The inclusion of specific clinical data, imaging findings, and postoperative results strengthens the report's credibility and informative value.
Recommendations for Modifications:
- Comparative Analysis: Including a comparative study or more extensive literature review on E-vac versus traditional methods could provide a stronger argument for the efficacy of E-vac. Also in the introduction, the procedure needs to be mentioned and described since the authors make no mention of this procedure but only of the classical ones.
- The authors mention that this is the first time this procedure was used. In their hospital? At the national level? These details need to be mentioned.
- The inflammatory markers need to be detailed in what way are linked to the procedure. Is there a trend that needs to be followed in the presented context? The graph and the values are not about this case which seems to be presenting a surgical procedure.
- The authors mention in their conclusion about septic shock, yet this is not mentioned in the case presentation. Septic shock has a specific treatment and definition with hemodynamic instability and vasopressor usage. Were this present in the presented patient? Was the patient admitted to the ICU? What about the widely used SOFA score or qSOFA calculated? Also, from the biomarkers panel the authors presented the lactate is not shown. In septic shock, lactate is a key biomarker.
- Long-Term Follow-Up Data: Adding information on the patient's long-term outcomes post-discharge would enhance the understanding of E-vac's effectiveness over time.
- Broader Context: Expanding the discussion to include a wider range of cases or a meta-analysis of similar cases could offer a more comprehensive view of E-vac's applicability and success rate.
- Limitations and Challenges: A more detailed discussion on the potential limitations, challenges, or complications of E-vac therapy would provide a balanced view and assist in clinical decision-making.
- Some images/ diagrams with the procedure itself need to be included.
- Future Research Directions: Suggestions for future research, such as randomized controlled trials or long-term studies, would be beneficial in guiding the next steps in this area of medical practice.
Author Response
Thank you for your appreciation and at the same time for taking the time to read our article. We have tried to respond to each sugestion.

Reviewer 3 Report
Comments and Suggestions for Authors
First of all, congratulations to the authors for the work done in writing the article.
Considering the exhaustive documentation of pathogenesis, symptomatology and investigations used in the management of perforated duodenal ulcer associated complications (both through open and minimally invasive approach) of complications, I recommend the authors to edit this article as a case presentation and narrative review of the literature( edit the title of the article as well).
Comments on the Quality of English Language
An additional correction of medical English is also necessary.
Author Response
Thank you for your appreciation and at the same time for taking the time to read our article.

Round 2
Reviewer 2 Report
Comments and Suggestions for Authors
The authors improved the quality of the article with the modifications included in the manuscript.